# Almost Surely Stable Deep Dynamics

**Nathan P. Lawrence**
Department of Mathematics
University of British Columbia
lawrence@math.ubc.ca

**Philip D. Loewen**
Department of Mathematics
University of British Columbia
loew@math.ubc.ca

**Michael G. Forbes**
Honeywell Process Solutions
michael.forbes@honeywell.com

**Johan U. Backström**
Backstrom Systems Engineering Ltd.
johan.u.backstrom@gmail.com

**R. Bhushan Gopaluni**
Department of Chemical and Biological Engineering
University of British Columbia
bhushan.gopaluni@ubc.ca

## Abstract

We introduce a method for learning provably stable deep neural network based dynamic models from observed data. Specifically, we consider discrete-time stochastic dynamic models, as they are of particular interest in practical applications such as estimation and control. However, these aspects exacerbate the challenge of guaranteeing stability. Our method works by embedding a Lyapunov neural network into the dynamic model, thereby inherently satisfying the stability criterion. To this end, we propose two approaches and apply them in both the deterministic and stochastic settings: one exploits convexity of the Lyapunov function, while the other enforces stability through an implicit output layer. We demonstrate the utility of each approach through numerical examples.

## 1 Introduction

Stability is a critical requirement in the design of physical systems. White-box models based on first principles can explicitly account for stability in their design. On the other hand, deep neural networks (DNNs) are flexible function approximators, well suited for modeling complicated dynamics. However, their black-box design makes both physical interpretation and stability analysis challenging.

This paper focuses on the construction of provably stable DNN-based dynamic models. These models are amenable to standard deep learning architectures and training practices, while retaining the asymptotic behavior of the underlying dynamics. Specifically, we focus on stochastic systems whose state $\boldsymbol{x}_t \in \mathbb{R}^n$ evolves in discrete time as follows:

$$\boldsymbol{x}_{t+1} = f(\boldsymbol{x}_t, \boldsymbol{\omega}_{t+1}), \quad t \in \mathbb{N}_0, \tag{1}$$

where $\boldsymbol{\omega}_t \in \mathbb{R}^d$ is a stochastic process. Although real physical systems typically evolve in continuous time, the periodic sampling of measurement and control signals in digital systems give great practical interest to discrete-time analysis. Moreover, noise often plays a prominent role in the underlying dynamics, making it an important feature to consider in the stability analysis. Our strategy starts from the philosophy proposed by Manek and Kolter [30]: It is easier to construct a stable dynamic model by simultaneously training a suitable Lyapunov function, than it is to separately verify stability for a trained model *a posteriori*. In this work, we propose two methods for guaranteeing stability of

deterministic discrete-time dynamic models: we first exploit convexity of a Lyapunov function given by a neural network, and then propose a general approach using an implicit output layer. We then show how to extend our framework from the deterministic case to the stochastic case.

## 2 Background

For brevity, we summarize basic stability results for stochastic systems, as the deterministic analogs can be readily inferred, for example, through discarding the expectation operator in Theorem 2.1. See, for example, [23, 24, 10] for precise statements of the deterministic results. Throughout this paper, $\boldsymbol{x} = \boldsymbol{0}$ is assumed to be an equilibrium point.

**Definition 2.1** (Stochastic stability [27–29]). In system (1), the origin is said to be:

1. *Stable in probability* if for each $\epsilon > 0$ we have
$$\lim_{\boldsymbol{x}_0 \to \boldsymbol{0}} \mathbb{P}\left[\sup_{t \in \mathbb{N}_0} \|\boldsymbol{x}_t\| > \epsilon\right] = 0.$$

2. *Asymptotically stable in probability* if it is stable in probability and, for each $\boldsymbol{x}_0 \in \mathbb{R}^n$,
$$\mathbb{P}\left[\lim_{t \to \infty} \|\boldsymbol{x}_t\| = 0\right] = 1.$$

3. *Almost surely (a.s) asymptotically stable* if we have
$$\mathbb{P}\left[\lim_{\boldsymbol{x}_0 \to \boldsymbol{0}} \sup_{t \in \mathbb{N}_0} \|\boldsymbol{x}_t\| = 0\right] = 1$$

   and for any $\boldsymbol{x}_0 \in \mathbb{R}^n$, all sample paths $\boldsymbol{x}_t \in \mathbb{R}^n$ converge to to the origin almost surely.

4. $m^{th}$ *mean stable* if
$$\lim_{\boldsymbol{x}_0 \to 0} \mathbb{E}\left[\|\boldsymbol{x}_t\|_m^m\right] = 0.$$

Almost sure stability is the direct analog of deterministic stability, as it simply asserts each sample path is stable a.s. The above definitions for asymptotic stability can be strengthened to *exponential stability* (in probability or almost surely) by replacing the convergence of sample trajectories $\boldsymbol{x}_t$ with convergence of $\eta^t \boldsymbol{x}_t$, where $\eta > 1$ is a fixed constant.

Lyapunov stability theory has been adapted to many contexts and is a keystone for analyzing nonlinear systems. Although it was developed to treat deterministic, continuous-time systems, the basic intuition from this setting can be applied to the stochastic and/or discrete-time settings as well. The quantitative difference between the continuous-time and discrete-time cases is the use of an infinitesimal operator, namely, the Lie derivative. Lyapunov stability for discrete-time (stochastic) systems simply checks for a sufficient (expected) decrease in the Lyapunov function between time steps. Throughout this paper, in both deterministic and stochastic settings, we use $V$ to refer to a (candidate) Lyapunov function. The standard hypotheses concerning $V$ are as follows:

1. $V \colon \mathbb{R}^n \to \mathbb{R}$ is continuous
2. $V(\boldsymbol{x}) > 0$ for all $\boldsymbol{x} \neq \boldsymbol{0}$, and $V(\boldsymbol{0}) = 0$
3. There exists a continuous, strictly increasing function $\varphi \colon [0, \infty) \to [0, \infty)$ such that $V(\boldsymbol{x}) \geq \varphi(\|\boldsymbol{x}\|)$ for all $\boldsymbol{x} \in \mathbb{R}^n$
4. $V(\boldsymbol{x}) \to \infty$ as $\|\boldsymbol{x}\| \to \infty$

The Lyapunov stability theorems provide sufficient conditions for stability. It is worth noting that in the stochastic case, sufficiency is achieved by showing convergence in expectation of $V$ (rather than in probability) and also by assuming the process is Markovian.

**Theorem 2.1** (**Lyapunov stability** [29, 34]). *Consider the system in Eq. (1). Let $V \colon \mathbb{R}^n \to \mathbb{R}$ be a continuous positive-definite function that satisfies $c_1 \|\boldsymbol{x}\|^a \leq V(\boldsymbol{x}) \leq c_2 \|\boldsymbol{x}\|^a$ for some $c_1, c_2, a > 0$.*

*Let $\boldsymbol{x}_0, \boldsymbol{x}_1, \boldsymbol{x}_2, \ldots$ be a Markov process generated by Eq. (1). Assume there is a fixed $0 < \alpha < 1$ such that for all $t \in \mathbb{N}_0$*

$$\mathbb{E}\left[V(\boldsymbol{x}_{t+1})|\boldsymbol{x}_t\right] - V(\boldsymbol{x}_t) \leq -\alpha V(\boldsymbol{x}_t) \quad a.s. \tag{2}$$

*Then the origin is globally exponentially stable a.s.; if the left hand side of Eq. (2) is only strictly negative then the origin is globally asymptotically stable in probability.*

**Lyapunov neural networks.** There have been a couple of proposed neural network architectures for Lyapunov functions. Richards et al. [35] propose the structure $V(\boldsymbol{x}) = \phi(\boldsymbol{x})^T \phi(\boldsymbol{x})$, where $\phi$ is a DNN whose weights are arranged such that $V$ is positive-definite. We refer to this structure as a Lyapunov neural network (LNN). Manek and Kolter [30] utilize input-convex neural networks (ICNNs) [3, 16] with minor modifications to define a valid Lyapunov function. A LNN can also be made convex through the same arrangement of weights used in ICNNs. In either case, we add a small term $\epsilon \|\boldsymbol{x}\|^2$ to the Lyapunov function in order to satisfy the lower-bounded property for $V$ described above. In this paper, we simply distinguish between Lyapunov functions based on convexity with the understanding that these architectures satisfy the conditions described above and can therefore be used in simulation examples in section 6.

## 3   Learning stable deterministic dynamics

In this section we present a couple of methods for constructing provably stable deterministic discrete-time dynamic models (i.e., $f$ has no $\boldsymbol{\omega}_t$-dependence in Eq. (1)). We first consider stability under convex Lyapunov functions, then generalize to a non-convex setting. In the next section we extend these results to stochastic models. Throughout this paper we use $\hat{f}$ to refer to a nominal DNN model, while $f$ refers to a stable DNN model derived from the nominal model.

### 3.1   Convex Lyapunov functions

Consider a discrete-time system of the form

$$\boldsymbol{x}_{t+1} = f(\boldsymbol{x}_t). \tag{3}$$

Our goal is to construct a DNN representation of $f$ with global stability guarantees about the origin. First, for simplicity, we define $\beta = 1 - \alpha \in (0, 1)$, where $\alpha$ is a fixed parameter as in Theorem 2.1. Given a Lyapunov function $V$ satisfying the conditions from section 2, and a nominal model $\hat{f}$, we define the following dynamics:

$$
\begin{aligned}
\boldsymbol{x}_{t+1} &= f(\boldsymbol{x}_t) \\
&\equiv \begin{cases} \hat{f}(\boldsymbol{x}_t) & \text{if } V(\hat{f}(\boldsymbol{x}_t)) \leq \beta V(\boldsymbol{x}_t) \\ \hat{f}(\boldsymbol{x}_t)\left(\frac{\beta V(\boldsymbol{x}_t)}{V(\hat{f}(\boldsymbol{x}_t))}\right) & \text{otherwise} \end{cases} \\
&= \gamma \hat{f}(\boldsymbol{x}_t), \qquad \text{where } \gamma = \gamma(\boldsymbol{x}_t) = \frac{\beta V(\boldsymbol{x}_t) - \texttt{ReLU}\big(\beta V(\boldsymbol{x}_t) - V(\hat{f}(\boldsymbol{x}_t))\big)}{V(\hat{f}(\boldsymbol{x}_t))}.
\end{aligned} \tag{4}
$$

A geometric interpretation of Eq. (4) is shown in Figure 1. It is worth noting that the entire model defined above is used for training, not just $\hat{f}$. Therefore, the underlying objective is to optimize $\hat{f}$ and $V$ subject to the stability condition imposed by Eq. (4). The stability proof only requires convexity of $V$ and the deterministic version of Theorem 2.1.

**Proposition 3.1** (**Stability of deterministic systems**). *Let $V$ be a convex candidate Lyapunov function as described in Theorem 2.1. Then the origin is globally exponentially stable for the dynamics given by Eq. (4).*

*Proof.* Fix $\beta \in (0, 1)$ and take $\gamma = \gamma(\boldsymbol{x}_t)$ from Eq. (4). If $V(\hat{f}(\boldsymbol{x}_t)) \leq \beta V(\boldsymbol{x}_t)$ we have $\gamma = 1$. Otherwise, $\gamma \in (0, 1)$. In either case, due to convexity of $V$ and the property $V(\boldsymbol{0}) = 0$, we have

$$V(\boldsymbol{x}_{t+1}) = V(\gamma \hat{f}(\boldsymbol{x}_t)) \tag{5}$$
$$\leq \gamma V(\hat{f}(\boldsymbol{x}_t)) \tag{6}$$
$$\leq \beta V(\boldsymbol{x}_t) \tag{7}$$
$$\iff V(\boldsymbol{x}_{t+1}) - V(\boldsymbol{x}_t) \leq -\alpha V(\boldsymbol{x}_t) \tag{8}$$

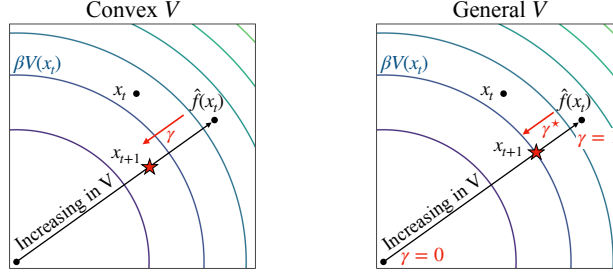

Figure 1: The intuition for our approach is to correct the nominal model $\hat{f}$ through scaling. (Left) $\gamma$ can be written in closed-form when $V$ is convex; (Right) In the general case, $\gamma^\star$ is written implicitly as the solution to a root-finding problem.

where $\alpha = 1 - \beta \in (0, 1)$. Therefore, the dynamics given by Eq. (4) are globally exponentially stable according to the deterministic analog of Theorem 2.1. □

Although it is not assumed in the definition of a Lyapunov function, convexity is a useful property due to the closed-form expression for $\gamma$ above. However, the scaling term $\gamma$ may be too restricted, as it relies explicitly on the convexity of $V$. We therefore propose a more general method that implicitly defines such $\gamma$.

### 3.2   Non-convex Lyapunov functions

In this section, $V$ is not assumed to be convex. For simplicity of exposition, we assume $\boldsymbol{x}_t \neq \boldsymbol{0}$.

The underlying strategy here is similar to that of the convex case: If a state transition using the nominal model $\hat{f}$ produces sufficient decrease in $V$, no intervention is required. Otherwise, since we cannot describe a suitable $\gamma$ in closed form, we seek a new state as follows:

$$\text{Find} \quad \boldsymbol{x}_{t+1}^\star \in \mathbb{R}^n \quad \text{such that} \quad V(\boldsymbol{x}_{t+1}^\star) - \beta V(\boldsymbol{x}_t) = 0 \tag{9}$$

Note a solution $\boldsymbol{x}_{t+1}^\star$ exists because $V$ is continuous and radially unbounded. Generally, the problem posed by (9) is a nonlinear $n$-dimensional root-finding problem whose solution is not unique. However, we can make (9) more tractable (both for prediction and training) by reducing it to a 1-dimensional root-finding problem:

$$\text{Find} \quad \gamma^\star \in \mathbb{R} \quad \text{such that} \quad V(\gamma^\star \hat{f}(\boldsymbol{x}_t)) - \beta V(\boldsymbol{x}_t) = 0 \tag{10}$$

It is worth noting that problem (10) is a generalization of Eq. (4) and is therefore state dependent; that is, $\gamma^\star = \gamma^\star(\boldsymbol{x}_t)$. Interestingly, we can solve (10) with any root-finding algorithm and it will not affect the training procedure, which we discuss later in this section. We use a robust hybrid between Newton's method and the bisection method. If $V$ is not convex, Newton's method is not guaranteed to solve problems (9) or (10) from an arbitrary initial value. However, by observing that $\gamma^\star \in (0, 1)$ whenever $V(\hat{f}(\boldsymbol{x}_t)) - \beta V(\boldsymbol{x}_t) > 0$, we can simply use the bisection method starting at $\gamma^{(0)} = 1$. Indeed, we have $V(\hat{f}(\boldsymbol{x}_t)) - \beta V(\boldsymbol{x}_t) > 0$ and $V(\boldsymbol{0}) - \beta V(\boldsymbol{x}_t) < 0$, so the existence of a solution is guaranteed by the intermediate value theorem. This procedure is illustrated in Figure 1. Of course, Newton's method is preferred. Therefore, if the Newton iteration takes the iterate $\gamma^{(i)}$ outside $[0, 1]$, then we discard this update and instead apply the bisection update, also constricting the interval $[0, 1]$ accordingly. Continuing in this fashion, we are guaranteed to find $\gamma^\star$ at most as fast as Newton's method. In summary, $\gamma^\star = \gamma^\star(\boldsymbol{x}_t)$ from (10) can be used in place of $\gamma$ from section 3.1. Concretely, we write the dynamic model as:

$$\boldsymbol{x}_{t+1} = f(\boldsymbol{x}_t)$$
$$\equiv \begin{cases} \hat{f}(\boldsymbol{x}_t) & \text{if } V(\hat{f}(\boldsymbol{x}_t)) \leq \beta V(\boldsymbol{x}_t) \\ \gamma^\star \hat{f}(\boldsymbol{x}_t) & \text{otherwise} \end{cases} \tag{11}$$

In the following theorem, we address the stability and continuity of the model given by Eq. (11). Simply put, the implicit model (11) inherits continuity through the nominal model $\hat{f}$ and Lyapunov

function $V$ via the implicit function theorem [37]. That is, the parameter $\gamma^\star$ varies continuously, even in regions of the state space in which both cases of the piece-wise rule in Eq. (11) are active. While the proof is fairly straightforward, it requires some care, so we provide the details in Appendix A. Finally, we note that in addition to the assumptions about $V$ from section 2, we assume $V$ is monotonically increasing in all directions from the origin and is continuously differentiable. These conditions are readily satisfied with the architectures for $V$ described in section 2.

**Theorem 3.1** (**Stability and continuity of implicit dynamics**). *Let $\hat{f}\colon \mathbb{R}^n \to \mathbb{R}^n$ be a nominal dynamic model and $V\colon \mathbb{R}^n \to \mathbb{R}$ be a candidate Lyapunov function. Assume $\hat{f}$ and $V$ are continuously differentiable. Further, assume for each fixed $\boldsymbol{x} \in \mathbb{R}^n$ that the function $h\colon \mathbb{R} \to \mathbb{R}$ given by $h(\gamma) = V(\gamma \hat{f}(\boldsymbol{x}))$ satisfies $h' > 0$. Then the dynamics defined by Eq. (11) are globally exponentially stable. Moreover, the model $f$ is locally Lipschitz continuous.*

*Proof.* Please see Appendix A for a detailed proof. $\square$

**Training implicit dynamic models.** Our stable implicit model falls into the class of implicit layers in deep learning [43, 4, 18, 1, 2]. This means that part of a DNN is not defined with the standard feed-forward structure, but rather an implicit statement describing the next layer. As such, the implicit function theorem can be used to train the model (11). In particular, problem (10) seeks a zero of the function $g(\gamma) = V(\gamma \hat{f}(\boldsymbol{x})) - \beta V(\boldsymbol{x})$, which has an invertible (nonzero) derivative at $\gamma^\star$. The backpropagation equations then follow by the chain rule.

We can also capitalize on the recent insights developed around deep equilibrium models (DEQs) [4]. The basic idea behind training a DEQ is to backpropagate through a fixed-point equation rather than through the entire sequence of steps leading to the fixed-point. Concretely, if $F(\gamma) = \gamma - g(\gamma)/g'(\gamma)$ is the standard scalar Newton iteration, then (10) is equivalently a fixed-point problem in $F$. We can therefore backpropagate through the fixed point given by the Newton iteration. Notably, this approach can still incorporate the bisection method, as backpropagation relies only on the end result $\gamma^\star$. This approach simplifies the implementation of training the implicit dynamic model and is our preferred method. In particular, since $F$ is in terms of both $\hat{f}$ and $V$, automatic differentiation tools enable streamlined parameter updates through use of $F$. For completeness, we give the corresponding backpropagation equations for these approaches in Appendix B.

## 4 Stochastic systems

We now extend our results from the deterministic setting to the stochastic setting. We start with the main result, then discuss its practical implementation.

**Mixture Density Networks.** Mixture density networks (MDNs) provide a simple and general method for modeling conditional densities [8, 20, 19, 42]. Concretely, we consider the form

$$p\left(\boldsymbol{x}_{t+1}|\boldsymbol{x}_t\right) = \sum_{i=1}^{k} \pi_i(\boldsymbol{x}_t)\phi_i(\boldsymbol{x}_{t+1}|\boldsymbol{x}_t), \tag{12}$$

where each $\phi_i$ is a kernel function (usually Gaussian) and the mixing coefficients $\pi_i$ are nonnegative and sum to $1$. The parameters for each kernel function and the respective mixing coefficients are the outputs of a DNN. Therefore, MDNs are appealing for our purposes of modeling stochastic dynamics because they are compatible with any DNN and a closed-form of the underlying mean and covariance is always available. Moreover, MDNs can be trained by minimizing the negative log-likelihood via standard backpropagation through the model. These are useful properties for adapting our methods from the deterministic case. Other stochastic models such as stochastic feed-forward neural networks [38] or Bayes by backprop [9] have a higher capacity for complex densities but lack some of these attributes of MDNs.

In the following theorem, *conditional mean dynamics* refers to the sequence of means $\boldsymbol{\mu}_1, \boldsymbol{\mu}_2, \ldots$ of Eq. (12), namely:

$$\boldsymbol{\mu}_{t+1} = \sum_{i=1}^{k} \pi_i(\boldsymbol{x}_t)\hat{\boldsymbol{\mu}}_i(\boldsymbol{x}_t), \tag{13}$$

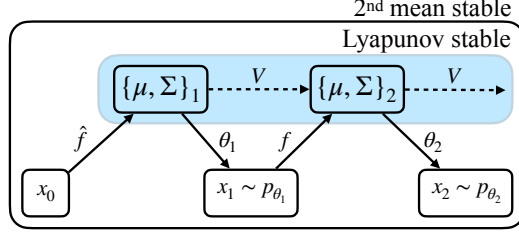

Figure 2: We impose stability on the conditional mean and variance dynamics produced by a MDP by means that ensure stability of the stochastic process $\boldsymbol{x}_t$. The dashed lines indicate a 'target' produced by $V$ based on the previous mean/covariance $\{\boldsymbol{\mu}, \Sigma\}_t$; here $\theta_t$ are the mixture parameters. The shaded region disentangles the stability of the mean/covariance dynamics from that of the state dynamics.

where $\hat{\boldsymbol{\mu}}_i \in \mathbb{R}^n$ is the conditional mean for mixture $i$. Similarly, we refer to the conditional covariance matrices of Eq. (12) as $\Sigma_t$ for each $t$. Note that these are stochastic processes as they depend on states $\boldsymbol{x}_t$. The following result shows that stochastic stability can be characterized in terms of conditional mean dynamics and the conditional covariance matrices of a MDN.

**Theorem 4.1** (**Stable stochastic dynamic models**). *Let $f \colon \mathbb{R}^n \to \mathbb{R}^\ell$ be a MDN model ($\ell$ is proportional to $n$ and the number of mixtures) and $V \colon \mathbb{R}^n \to \mathbb{R}$ be a candidate Lyapunov function satisfying the conditions of Theorem 3.1. Assume $c_1 \|\boldsymbol{x}\|^2 \le V(\boldsymbol{x})$ for some $c_1 > 0$. Let $\boldsymbol{\mu}_{t+1}$ denote the conditional mean dynamics and $\Sigma_{t+1}$ denote the conditional covariances. Assume the conditional mean dynamics are stable in probability according to $V$. If the maximum eigenvalue of $\Sigma_{t+1}$ is proportional to $V(\boldsymbol{\mu}_{t+1})$ for all $t$, then the stochastic system generated by $\hat{f}$ is $2^{nd}$ mean stable.*

*Proof.* Note that for all $t \in \mathbb{N}_0$

$$\mathbb{E}\left[\|\boldsymbol{x}_{t+1}\|^2 \Big| \boldsymbol{x}_t\right] = \text{Trace}[\Sigma_{t+1}] + \|\boldsymbol{\mu}_{t+1}\|^2 \le \text{Trace}[\Sigma_{t+1}] + \frac{1}{c_1} V(\boldsymbol{\mu}_{t+1}) = \mathcal{O}\left(V(\boldsymbol{\mu}_{t+1})\right)$$

because $V$ is lower bounded by $c_1 \|\boldsymbol{x}\|^2$ and because the trace of $\Sigma_{t+1}$ is the sum of its eigenvalues.

Now, fix $\epsilon > 0$. Let $c$ be a constant that achieves the above upper bound. By continuity of $\hat{f}$ and $V$, let $\delta > 0$ be such that we have $cV(\boldsymbol{\mu}_1) < \epsilon$ whenever $\|\boldsymbol{x}_0\| < \delta$.

We then have

$$\mathbb{E}\left[\|\boldsymbol{x}_1\|^2\right] = \mathbb{E}\left[\mathbb{E}\left[\|\boldsymbol{x}_1\|^2 \Big| \boldsymbol{x}_0\right]\right] \tag{14}$$

$$\le c\mathbb{E}\left[V(\boldsymbol{\mu}_1)\right] < \epsilon \tag{15}$$

Since we have that $V(\boldsymbol{\mu}_{t+1}) \le V(\boldsymbol{\mu}_t)$ a.s. for all $t \in \mathbb{N}_0$ it follows that Eq. (15) holds for all $t \in \mathbb{N}_0$ and all trajectories such that $\|\boldsymbol{x}_0\| < \delta$. Therefore, the stochastic system generated by $\hat{f}$ is $2^{nd}$ mean stable. $\square$

**Remark 4.1.** The above assumptions can be relaxed to only require continuity of $f$ and $V$, and convexity of $V$ if the techniques from section 3.1 are employed on the conditional means instead of the implicit dynamics approach.

**Stable mean dynamics.** The intuition behind Theorem 4.1 is to differentiate between the trajectory of the conditional means $\boldsymbol{\mu}_{t+1}$ and that of the states $\boldsymbol{x}_t$. In particular, each sample path of the conditional means can be constrained to decrease in $V$ using the tools from section 3. This is because $\gamma$ (from section 3.1) and $\gamma^\star$ (from section 3.2) are explicitly designed to bring new 'states' $\boldsymbol{\mu}_{t+1}$ to a desired level set, such as $V_{\text{target}} = \beta V(\boldsymbol{\mu}_t)$. In this way, we impose $V(\boldsymbol{\mu}_{t+1}) \le \beta V(\boldsymbol{\mu}_t)$ a.s. for all $t$, and consequently, $\mathbb{P}\left[\lim_{t \to \infty} \|\boldsymbol{\mu}_t\| = 0\right] = 1$ due to Theorem 2.1, as each sample path of $\boldsymbol{\mu}_{t+1}$ produces sufficient stepwise decreases in $V$. It is worth noting that the expectation operator in Theorem 2.1 is intractable over a general $V$, and therefore motivates our approach of unifying Lyapunov stability of the conditional means with the structure of a MDN to ultimately arrive at $2^{nd}$ mean stability. A schematic of this idea is shown in Figure 2.

Stability of the means does not necessarily imply stability of the stochastic system, which is why we also require the covariance goes to zero. Though other conditions are possible, Theorem 4.1

prescribes the simple condition that the eigenvalues of $\Sigma_{t+1}$ must vanish with $\boldsymbol{\mu}_{t+1}$. This can be achieved by restricting the covariances of each mixture to be diagonal, then bounding them and scaling, for example, by $\|\boldsymbol{\mu}_{t+1}\|$ or $V(\boldsymbol{\mu}_{t+1})$. Requiring the covariances to be diagonal in a mixture model is not a significant drawback, as more mixtures may be used [8].

## 5  Related work

Our work is most similar in spirit to that of Manek and Kolter [30]. However, their proposed approach is for deterministic, continuous-time systems, whereas this paper is concerned with learning from noisy discrete measurements $\boldsymbol{x}_t, \boldsymbol{x}_{t+1}, \ldots$ (rather than observations of the functions $\boldsymbol{x}(\cdot)$ and $\dot{\boldsymbol{x}}(\cdot)$). Discrete-time systems with stochastic elements require completely different analysis. Lyapunov stability theory has been deployed in several other recent machine learning and reinforcement learning works. Richards et al. [35] introduce a general neural network structure for representing Lyapunov functions. The approach is used to estimate the largest region of attraction for a fixed deterministic, discrete-time system. Umlauft and Hirche [39] consider the stability of nonlinear stochastic models under certain state transition distributions. However, their approach is constrained to provably stable stochastic dynamics under a quadratic Lyapunov function. Khansari-Zadeh and Billard [25] consider Gaussian mixture models for learning continuous-time dynamical systems but only enforce stability of the means. Wang et al. [40] develop dynamical models in which the latent dynamics and observations follow Gaussian Processes; stability analysis is later given by Beckers and Hirche [5, 6]. In reinforcement learning, [7, 17, 13] utilize Lyapunov stability to perform safe policy updates within an estimated region of attraction.

Stability analysis has also been incorporated into the design, training, and interpretation of neural networks. For example, Haber and Ruthotto [21], Chang et al. [12] view a residual network as a discretization of an ordinary differential equation, leading to improved sample efficiency in image classification tasks due to the well-posedness and stability of the underlying dynamics. In the same vein, Chen et al. [15], Chen and Duvenaud [14] directly parameterize the time derivative of the hidden state dynamics and utilize a numerical ODE solver for predictions, then extend these ideas to stochastic differential equations. Recurrent models also describe dynamical systems and therefore have been studied through this lens, for example, by Miller and Hardt [31], Bonassi et al. [11]. By extension, the optimality of identified models through (stochastic) gradient descent on linear and nonlinear dynamics has been studied [22, 32].

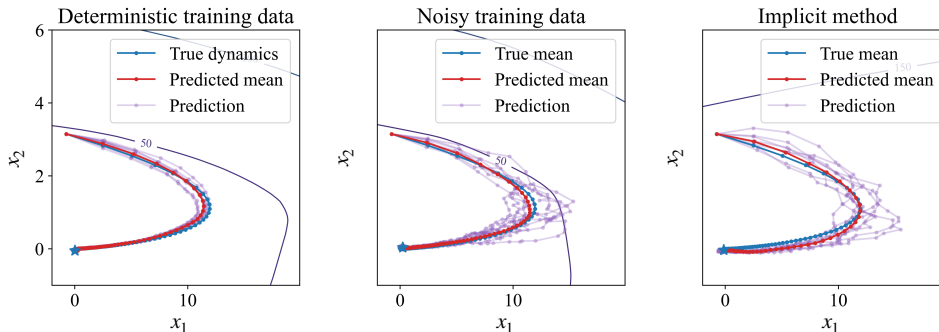

Figure 3: This example illustrates all of the methods developed in this paper. (Left) Sample trajectories after training a stable MDN with data from a deterministic system; (Middle) Sample trajectories after training with noisy data; (Right) Same as the middle, but with the implicit stability method to define the dynamics.

## 6  Experiments

The code for our methods is available here: `https://github.com/NPLawrence/stochastic_dynamics`.

We first show a toy example with linear systems to illustrate all the methods presented in this paper. We then give numerical results for nonlinear systems, both deterministic and stochastic. We use a

fully connected feedforward neural network for both $\hat{f}$ and $V$. Further details about the experiments and models can be found in Appendix D. We also give an example dealing with a chaotic system in Appendix C. Although the examples here deal with low-dimensional state spaces for convenient visualizations, we note that our method is not restricted to this setting, as the dynamic model is based on DNNs and thus can take any state dimension.

**Example 1 (A linear system).** It is well-known that a quadratic Lyapunov function can be obtained for a stable deterministic linear system by solving the Lyapunov equation (Algebraic Riccati equation for dynamics without inputs). A similar statement holds for stochastic linear systems of the form

$$\boldsymbol{x}_{t+1} = A\boldsymbol{x}_t + B\boldsymbol{x}_t\omega_t, \quad \text{where } \omega_t \sim \mathcal{N}(0,1). \tag{16}$$

In particular, if for any positive definite $Q$ we can solve $A^T P A + B^T P B - P + Q = 0$ for some positive definite $P$, then $V(\boldsymbol{x}) = \boldsymbol{x}^T P \boldsymbol{x}$ can be used to certify stochastic stability of the system (16). It is then clear that for a linear system there are many valid Lyapunov functions. This justifies their use and design in constructing stable DNN models both in deterministic and stochastic settings.

Results are shown in Figure 3 and correspond to the matrix

$$A = \begin{bmatrix} 0.90 & 1 \\ 0 & 0.90 \end{bmatrix}$$

in Eq. (16). In our first experiment, we use training data from the system (16) in which there is no noise. As such, the MDN gives very small variance in its predictions. The predicted mean refers to the dynamics defined by feeding the means through the MDN as 'states' (i.e. no sampling). The next two plots show predictions corresponding to the system (16) with $B = 0.1$, where the last plot uses implicit dynamics.

**Example 2 (Non-convex Lyapunov neural network).** Consider the system

$$\begin{aligned} \dot{x} &= y \\ \dot{y} &= -y - \sin(x) - 2\mathtt{sat}(x+y), \end{aligned} \tag{17}$$

where $\mathtt{sat}(u) = u$ for $-1 < u < 1$ and $\mathtt{sat}(u) = u/|u|$ otherwise. It can be shown that the origin is globally asymptotically stable in the system (17), in part, by considering the nonquadratic Lyapunov function $V(x) = x^2 + 0.5y^2 + 1 - \cos(x)$ [24]. The trajectories in Figure 4 were computed using our methods with an ICNN-based Lyapunov function, a LNN, and a convex LNN that follows the ICNN construction. For the LNN case, we train the model using the implicit method, while the other two use the convexity-based method. It is worth noting that the system (17) is not the 'true' system in our experiment, rather the models are trained on a coarse discretization (time-step $h = 0.1$) of Eq. (17) via a fourth order Runge-Kutta method. Figure 4 shows sample trajectories corresponding to initial conditions not seen during training. Notably, both convexity-based simulations deviate from the true trajectory, whereas the implicit method is able to more accurately navigate the regions in the $x_1$-$x_2$ plane with fluctuations before descending toward the origin.

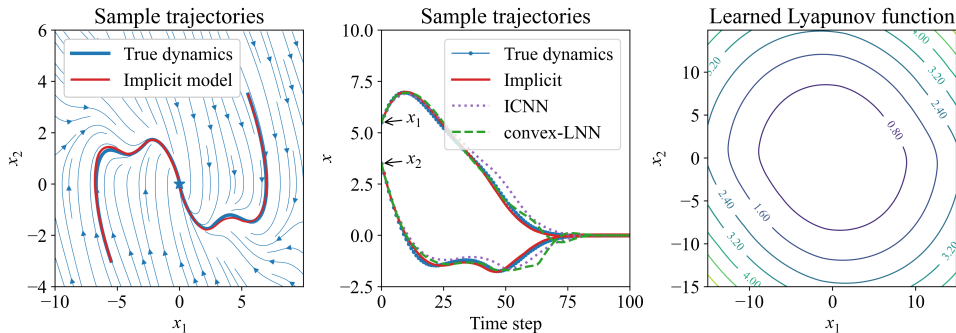

Figure 4: (Left) Continuous-time dynamics given with arrows underlying two bold discrete-time trajectories; (Middle) Sample trajectories corresponding to (non-)convex Lyapunov functions (Right) Learned $V$ for the implicit model method. Trajectories are discrete, but not dotted for clarity.

**Example 3 (Nonlinear stochastic differential equation).** Consider the stable nonlinear stochastic differential equation defined in terms of independent standard Brownian motions $B_1, B_2$ as follows

[41]:

$$dx_1 = \left( \frac{-x_1}{\sqrt{\|\boldsymbol{x}\|}} - x_1 + x_2 \right) dt + \sin(x_1)\, dB_1$$

$$dx_2 = \left( \frac{-x_2}{\sqrt{\|\boldsymbol{x}\|}} - \frac{10}{3}x_2 + x_1 \right) dt + x_2\, dB_2. \tag{18}$$

(Interpret $x_i / \sqrt{\|\boldsymbol{x}\|}$ as 0 at the origin.) This example illustrates the inherent stability of our stochastic model even when only partial trajectories are used for training. We use $k = 6$ mixtures and compare the performance of a convexity-based stable stochastic model against a standard MDN. In our experiment we discretize Eq. (18) using a second-order stochastic Runge-Kutta method with time-step $h = 0.05$ (see [36]). Tuples of the form $(\boldsymbol{x}_t, \boldsymbol{x}_{t+1})$ are used for training, so we use $\boldsymbol{x}_t$ in place of $\boldsymbol{\mu}_t$ (which is unavailable) with which we train $V$ through enforcing the condition $V(\boldsymbol{\mu}_{t+1}) \leq \beta V(\boldsymbol{x}_t)$. However, the same scheme cannot necessarily be applied to entire roll-outs while ensuring stability.

Figure 5 shows sample trajectories from both models as well as their performance. We show sample trajectories generated by each model alongside a representative sample from the true system. The smooth red lines show the mean dynamics (not the conditional means) as described in example 1. The performance plot shows the average negative log-likelihood (NLL) at each time step over 20 trajectories corresponding to initial values not seen during training. From Figure 5 we see the importance of stability as an inherent property of the dynamic model over a standard MDN.

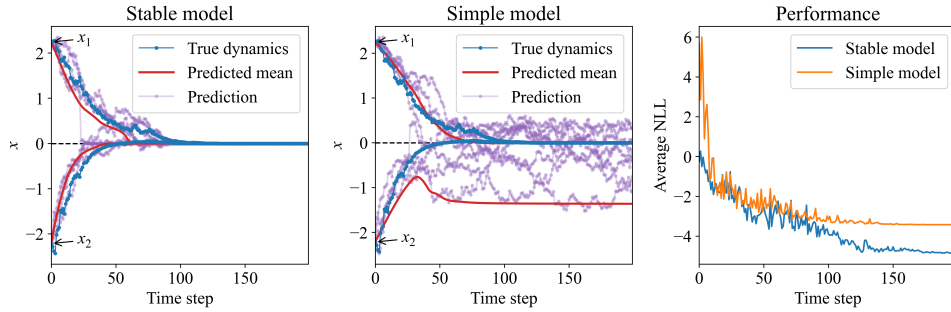

Figure 5: Sample paths of the system from example 3 and the learned model. The two initial values correspond to the two components of the state $(x_1, x_2) \in \mathbb{R}^2$.

## 7  Conclusion

We have developed a framework for constructing neural network dynamic models with provable global stability guarantees. We showed how convexity can be exploited to give a closed-form stable dynamic model, then extended this approach to implicitly-defined stable models. The latter case can be reduced to a one-dimensional root-finding problem, making a robust and cheap implementation straightforward. Finally, we leverage these methods to the stochastic setting in which stability guarantees are also given through the use of MDNs. A proof of concept of these methods was given on several systems of increasing complexity. The simplicity of our approach, combined with the expressive capacity of DNNs, makes it a pragmatic tool for modeling nonlinear dynamics from noisy state observations. Moreover, interesting avenues for future work include applications to control and reinforcement learning.

## Broader Impact

Stability goes hand in hand with safety. Therefore, stability considerations are crucial for the broad acceptance of DNNs in real-world industrial applications such as control, self-driving vehicles, or anaesthesia feedback, to name a few. To this end, industries or companies with sufficient computational and storage resources would benefit significantly through the use of such autonomous and interpretable technologies. However, this work mostly provides some theory and a proof of concept for stable DNNs in stochastic settings and as such does not pose a clear path nor an ethical quandary regarding such widespread control applications.

## Acknowledgments and Disclosure of Funding

We gratefully acknowledge the financial support of the Natural Sciences and Engineering Research Council of Canada (NSERC) and Honeywell Connected Plant.

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
