[Supplementary Material]

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

Figure 6: A schematic of the proof of Theorem A.1. $\varphi$ refers to a continuous function, derived from the implicit function theorem, taking $\hat{f}$ to states that decrease in $V$.

## A   Continuity of implicit dynamic models

**Theorem A.1 (Stability and continuity of implicit dynamics).** *Let $\hat{f} : \mathbb{R}^n \to \mathbb{R}^n$ be a nominal dynamic model and $V : \mathbb{R}^n \to \mathbb{R}$ be a candidate Lyapunov function. Assume $\hat{f}$ is locally Lipschitz continuous and $V$ is continuously differentiable. Further, assume for each fixed $\boldsymbol{x} \in \mathbb{R}^n$ that the function $h \colon \mathbb{R} \to \mathbb{R}$ given by $h(\gamma) = V(\gamma \hat{f}(\boldsymbol{x}))$ satisfies $h' > 0$. Then the dynamics defined by Eq. (11) are globally exponentially stable. Moreover, the model $f$ is locally Lipschitz continuous.*

*Proof.* For each $\boldsymbol{x}$ such that $V(\hat{f}(\boldsymbol{x})) > \beta V(\boldsymbol{x})$, there exists a unique solution to the equation

$$V(\gamma^\star \hat{f}(\boldsymbol{x})) - \beta V(\boldsymbol{x}) = 0 \tag{19}$$

because $V$ is strictly increasing in all directions from the origin and is radially unbounded. Therefore, the implicit based dynamic model is defined everywhere in $\mathbb{R}^n$. It is then clear, by construction, that the implicit approach yields exponentially stable discrete-time dynamics in the deterministic sense of Theorem 2.1. In practice, we find the root such that $\|V(\boldsymbol{x}_{t+1}^\star) - \beta V(\boldsymbol{x}_t)\| < \epsilon$, where $\epsilon > 0$ is a pre-defined tolerance. If the tolerance is set such that $\epsilon \leq V(\boldsymbol{x}_t)(1 - \beta)$ then the model is still stable (not necessarily exponentially) subject to small numerical error.

Now we show the implicit method is continuous. That is, close initial values find close roots. Geometrically, this is not surprising and follows from the implicit function theorem. Fix any positive target value $V_{\text{target}}$ (for instance, $V_{\text{target}} = \beta V(\boldsymbol{x})$ for a given $\boldsymbol{x}$). Let $\boldsymbol{x}^{(0)}$ be such that $V(\boldsymbol{x}^{(0)}) > V_{\text{target}}$. We are then interested in the following equation of $n + 1$ variables

$$V(\gamma \boldsymbol{x}^{(0)}) - V_{\text{target}} = 0. \tag{20}$$

From the above discussion we know there is a $\gamma^\star \in (0, 1)$ that satisfies Eq. (20). Recall $h$, as defined in our hypotheses, is non-stationary at $\gamma^\star$. Therefore, by the implicit function theorem, there exists some neighborhood $\mathcal{U}$ of $\boldsymbol{x}^{(0)}$ such that there is a continuously differentiable function $\varphi : \mathcal{U} \to \mathbb{R}$ satisfying $\varphi(\boldsymbol{x}^{(0)}) = \gamma^\star$ and

$$V(\varphi(\boldsymbol{x}^{(0)})\boldsymbol{x}^{(0)}) - V_{\text{target}} = 0 \tag{21}$$

for all $\boldsymbol{x}^{(0)} \in \mathcal{U}$. This establishes continuity in $\gamma^\star$ over a neighborhood of any initial iterate $\boldsymbol{x}^{(0)}$.

Now, fix any $\boldsymbol{x}$ such that the implicit method is needed ($\boldsymbol{x}$ does not decrease sufficiently in $V$). Let $\gamma^\star \hat{f}(\boldsymbol{x})$ be a solution satisfying Eq. (20) and define a neighborhood $\mathcal{U}_{\text{sol}}$ around $\gamma^\star \hat{f}(\boldsymbol{x})$. Let $\mathcal{U}_{\text{init}}$ be a neighborhood around $\hat{f}(\boldsymbol{x})$ as in the previous paragraph (to ensure continuity from $\mathcal{U}_{\text{init}}$ into $\mathcal{U}_{\text{sol}}$. By continuity of $\hat{f}$ and $V$, there is a neighborhood $\mathcal{U}$ in $\mathbb{R}^n$ such that $\hat{f}(\boldsymbol{x}) \in \mathcal{U}_{\text{init}}$ for all $\boldsymbol{x} \in \mathcal{U}$, namely, some neighborhood $\mathcal{U}_{\hat{f}} \subset \mathcal{U}_{\text{init}}$. Consequently, $\varphi(\hat{f}(\boldsymbol{x}))\hat{f}(\boldsymbol{x}) \in \mathcal{U}_{\text{sol}}$ for $\boldsymbol{x} \in \mathcal{U}$. That is, $\mathcal{U}$ is a neighborhood of the domain which maps into $\mathcal{U}_{\text{sol}}$. Moreover, since $\varphi$ is locally Lipschitz (because it is continuously differentiable), the implicit method is also locally Lipschitz. This follows by further restricting $\mathcal{U}_{\hat{f}}$ and $\mathcal{U}$ to smaller neighborhoods in which $\hat{f}$ and $\varphi$ satisfy the Lipschitz condition.

Finally, the entire dynamic model $f$ is locally Lipschitz continuous. Indeed, the above argument does not depends on the aforementioned restriction of the roots to $(0, 1)$. Instead, for any $\boldsymbol{x}$ we can perform the implicit method and retain the local Lipschitz continuity described above. Therefore, the entire dynamic model can be written as[1]

$$f(\boldsymbol{x}) = \texttt{sat}(\gamma^\star)\hat{f}(\boldsymbol{x}), \tag{22}$$

where $\gamma^\star > 0$.

$\square$

## B   Training implicit dynamic models

The following equations are only needed when the nominal model does not decrease sufficiently in $V$, otherwise standard backpropagation applies. The following are direct consequences of the implicit function theorem or implicit differentiation, for example, as in [43] or [4] respectively. In both cases presented below, we assume $\hat{f}$ and $V$ satisfy the hypotheses of Theorem A.1 and that $\mathcal{L} : \mathbb{R}^n \times \mathbb{R}^n \to \mathbb{R}$ is a differentiable loss function. Moreover, we define the scalar-valued function $g(\gamma) = V(\gamma \hat{f}(\boldsymbol{x})) - T(\boldsymbol{x})$, where $T$ defines a target value (e.g., $\beta V(\boldsymbol{x})$). Recall $\boldsymbol{x}_{t+1}^\star = \gamma^\star \hat{f}(\boldsymbol{x}_t)$.

**Direct calculation.**   The gradient of the loss $\mathcal{L}$ with respect to $(\cdot)$ is given by:

$$\frac{\partial \mathcal{L}}{\partial (\cdot)} = \frac{\partial \mathcal{L}}{\partial \boldsymbol{x}_{t+1}^\star} \frac{\partial \boldsymbol{x}_{t+1}^\star}{\partial (\cdot)}, \tag{23}$$

where

$$\frac{\partial \boldsymbol{x}_{t+1}^\star}{\partial (\cdot)} = \begin{cases} \gamma^\star I_{n \times n} - \dfrac{1}{\nabla V(\gamma^\star \hat{f}(\boldsymbol{x}))^T \hat{f}(\boldsymbol{x})} \hat{f}(\boldsymbol{x}) \frac{\partial g}{\partial \hat{f}}(\boldsymbol{x}) & \text{if} \quad (\cdot) = \hat{f}(\boldsymbol{x}) \\ \dfrac{1}{\nabla V(\gamma^\star \hat{f}(\boldsymbol{x}))^T \hat{f}(\boldsymbol{x})} \hat{f}(\boldsymbol{x}) & \text{if} \quad (\cdot) = T(\boldsymbol{x}). \end{cases}$$

**Fixed point approach.**   We use the notation $\gamma^{(i+1)} = F(\gamma^{(i)}; \boldsymbol{x}_t) \equiv \gamma^{(i)} - g(\gamma^{(i)})/g'(\gamma^{(i)})$ to denote the standard scalar Newton iteration. The superscripts denote the iteration. Therefore, $\gamma^{(i+1)} \in \mathbb{R}$, is a candidate scaling term for $\hat{f}(\boldsymbol{x}_t)$ following $\boldsymbol{x}_t$ at iteration $i + 1$ in the procedure.

Let $\gamma^\star = F(\gamma^\star; \boldsymbol{x}_t)$. Then the gradient of the loss with respect to $(\cdot)$ is given by:

$$\frac{\partial \mathcal{L}}{\partial (\cdot)} = \frac{\partial \mathcal{L}}{\partial \boldsymbol{x}_{t+1}^\star} \frac{\partial \boldsymbol{x}_{t+1}^\star}{\partial (\cdot)} \tag{24}$$

Since $\boldsymbol{x}_{t+1}^\star = \gamma^\star \hat{f}(\boldsymbol{x}_t)$, we only need to compute $\partial \gamma^\star / \partial (\cdot)$ then the rest follows by the product rule.

To this end, we have

$$\frac{\partial \gamma^\star}{\partial (\cdot)} = \frac{\partial F(\gamma^\star; \boldsymbol{x}_t)}{\partial (\cdot)} + \underbrace{\frac{\partial F(\gamma^\star; \boldsymbol{x}_t)}{\partial \gamma^\star}}_{0} \frac{\partial \gamma^\star}{\partial (\cdot)} \tag{25}$$

$$= \frac{\partial F(\gamma^\star; \boldsymbol{x}_t)}{\partial (\cdot)}. \tag{26}$$

## C   Experiment with chaotic systems

Before we show an example with the Lorenz attractor, we introduce a new model structure. The problem of ensuring stability of a dynamic model is closely related to the problem of minimizing the Lyapunov function $V$ through an iterative process. In particular, we require that the dynamic model traverses $V$ in a descent direction. In order to ensure this condition, we consider the case where an increment between time steps does not move in a descent direction:

$$\nabla V(\boldsymbol{x}_t)^T \big( \hat{f}(\boldsymbol{x}_t) - \boldsymbol{x}_t \big) > 0$$

at some time $t$. In this case, we can project the dynamics $\hat{f}(\boldsymbol{x}_t)$ onto the gradient $\nabla V(\boldsymbol{x}_t)$ to get dynamics normal to the gradient:

$$f(\boldsymbol{x}_t) \leftarrow \hat{f}(\boldsymbol{x}_t) - \nabla V(\boldsymbol{x})^T \big( \hat{f}(\boldsymbol{x}_t) - \boldsymbol{x}_t \big) \frac{\nabla V(\boldsymbol{x}_t)}{\|\nabla V(\boldsymbol{x}_t)\|^2}.$$

In closed form, we can write

$$\begin{aligned} \boldsymbol{x}_{t+1} &= f(\boldsymbol{x}_t) \\ &\equiv \begin{cases} \hat{f}(\boldsymbol{x}_t) & \text{if } \nabla V(\boldsymbol{x}_t)^T \big( \hat{f}(\boldsymbol{x}_t) - \boldsymbol{x}_t \big) \leq 0 \\ \hat{f}(\boldsymbol{x}_t) - \nabla V(\boldsymbol{x})^T \big( \hat{f}(\boldsymbol{x}_t) - \boldsymbol{x}_t \big) \frac{\nabla V(\boldsymbol{x}_t)}{\|\nabla V(\boldsymbol{x}_t)\|^2} & \text{otherwise} \end{cases} \\ &= \hat{f}(\boldsymbol{x}_t) - \texttt{ReLU}\big( \nabla V(\boldsymbol{x})^T \big( \hat{f}(\boldsymbol{x}_t) - \boldsymbol{x}_t \big) \big) \frac{\nabla V(\boldsymbol{x}_t)}{\|\nabla V(\boldsymbol{x}_t)\|^2}. \end{aligned} \tag{27}$$

This model does not guarantee stability, but can be utilized in an interesting way, as we demonstrate below.

**Example 4 (chaotic system).** The Lorenz attractor is a chaotic system, making it an interesting benchmark for DNN dynamic models. It's dynamics are given as follows:

$$\begin{aligned} \dot{x} &= \sigma\big(y - x\big) \\ \dot{y} &= x\big(\rho - z\big) - y \\ \dot{z} &= xy - \beta z. \end{aligned} \qquad (28)$$

Depending on its parameters, it may have a non-zero equilibrium or enter a limit cycle. For modeling a system with a non-zero equilibrium, we could use a priori knowledge of such a point $\boldsymbol{x}^\star$ then employ a variable shift $\boldsymbol{x}_{t+1} - \boldsymbol{x}^\star$. Instead, we consider a dynamic model with the integrating structure:

$$\boldsymbol{x}_{t+1} = \boldsymbol{x}_t + f(\boldsymbol{x}_t). \qquad (29)$$

In particular, $f$ is given by Eq. (27). Geometrically, this gives a state-dependent constraint on the direction and magnitude of the increment between time steps. This is useful for systems that are sensitive to small shifts in the state space.

In our experiments, we use a fourth order Runge-Kutta method to discretize Eq. (28). We use the standard parameters for the system: $\sigma = 10$, $\beta = 8/3$, $\rho = 28$. The models were trained on a single trajectory of 3,000 time steps (initial condition [1, 1, 1]) and the figures shown give their respective roll-outs for a slightly perturbed initial condition.

Figure 7: Side-by-side comparison of a 3000 time step roll-out between a bounded increments model and an unconstrained counterpart.

Figure 8: A view of each individual state for both models.

# D   ML Reproducibility

We summarize *additional* details that are not given in the main body of the paper.

## D.1   Models & Algorithms

In all experiments, $\hat{f}$ is a $n$-25-25-$\ell$ fully connected feedforward network, where $\ell$ is either $n$ or $2nk$, where $k$ is the number of mixtures in an MDN. For ease, a separate network is used to output the mixture coefficients. It is worth noting that $\hat{f}$ may be given by any parametric/differentiable representation of the form $\boldsymbol{x}_{t+1} = \hat{f}(\boldsymbol{x}_t, \boldsymbol{\omega}_{t+1})$. Therefore, other architectures such as convolutional neural networks and regularization techniques such as dropout are applicable. $V$ can be either an ICNN or Lyapunov neural network as described in section 2, in either case it uses a $n$-25-25-1 fully connected network. $V$ uses a custom activation proposed in [30] that gives a smooth approximation of ReLU. The term $\beta$ associated with the rate of decrease of $V$ was set to 0.99, as it only needs to be a value between 0 and 1, but closer to 1 gives more flexibility. The only component that requires some complexity analysis is the implicit dynamics method. For the root-finder we set the error tolerance to be 0.001. Since we combine Newton's method with the bisection method, this requires at most 10 bisection steps. Since the bisection method is a "back up", this would mean Newton's method was also executed for 10 steps but deviated from the current (or initial) interval given by the bisection method at each step. In the case $V$ is convex, Newton's method is guaranteed to converge (so, bisection method is not used) starting at $\gamma^{(0)} = 1$, since this value lies to the right of the zero of the increasing convex function $g(\gamma) = V(\gamma \hat{f}(\boldsymbol{x}_t)) - \beta V(\boldsymbol{x}_t)$. This is for time steps at which the root-finder is needed, otherwise we only execute the forward pass of $\hat{f}$.

## D.2   Datasets

The data can be generated using the given dynamical systems and Runge-Kutta schemes. In all experiments, we gather training data by recording tuples of the form $(\boldsymbol{x}_t, \boldsymbol{x}_{t+1})$ from trajectories corresponding to a grid ($14 \times 14$ equally spaced points in the interval $[-6, 6] \times [-6, 6]$) of initial values. The first two examples use trajectories of 40 time steps, while example 3 uses trajectories of 10 time steps. We did not pre-process the data because the models are describing a dynamical system. Models are evaluated based on average error (mean squared error or negative log-likelihood) across time steps over 20 trajectories corresponding to new initial values.

## D.3   Experimental Results

We implemented our models using PyTorch [33] and its default parameter initializations. Parameters are updated using Adam [26] to minimize the mean squared error or negative log-likelihood. We did not optimize the hyper-parameters. The default hyper-parameters for Adam were satisfactory for all experiments, but possibly required training for longer. We used the slightly larger learning rate 0.0025 and trained for $200 - 1000$ epochs. All experiments were ran on a laptop with a Quad-Core i7 processor and 16GB of RAM. Runtime was negligible for convexity based-models (as it is simply the forward/backward pass of $\hat{f}$ and $V$), but for the implicit method it is approximately 1.5-3 times slower depending on the level of accuracy in the root-finder.