[Reviews · NeurIPS 2020]

Review 1

Summary and Contributions: The authors propose a neural network architecture for learning stochastic dynamic systems, with the constraint that the learned dynamics are stable. This is achieved through a specialized final layer representing a Lyapunov function. The desired constraints on the Lyapunov function, sufficient to give stability, are enforced on this final layer, conferring stability to the learned dynamics. The approach is demonstrated on several simulated systems.

Strengths: Stability in learned dynamics is very important for applications in which forecasting plays some important role. In control applications, instability can lead to extreme control signals and catastrophic failure for example. For this reason the authors' work is significant and relevant to a number of communities.

Weaknesses: I see no major weaknesses. The authors' work is perhaps a reasonable step forward.

Correctness: The methodology appears correct.

Clarity: Overall the paper is reasonably clear. However it would benefit from a concise and complete description of the network architecture, perhaps in the beginning of Section 6 (since all relevant concepts have been introduced by this point).

Relation to Prior Work: The relationship with prior work is clearly described. The authors' describe differences with the related literature, but could do a slightly better job of describing the consequences of those differences. For example the work of Umlauft and Hirche is described (line 206) as pertaining to "certain state transitions." Is this a limiting factor in their work or not? How does these "certain state transitions" relate to the authors' work?

Reproducibility: Yes

Additional Feedback: I find the authors' rebuttal to be satisfactory. I am content to leave my score unchanged, but increase my confidence.


Review 2

Summary and Contributions: The authors propose a method to impose stability on learned stochastic discrete-time systems. The proposed method is in line with the idea of learning a neural Lyapunov function simultaneously, and they apply it to the discrete-time setting and also extend to the use of non-convex Lyapunov (candidate) functions. They also suggest using mixture density networks for modeling stochastic systems and show that the proposed framework can impose moment stability.

Strengths: The motivation and the method are technically sound and match well. As learning discrete-time stochastic systems is very frequently encountered in ML practices, this paper will be of great interest in the NeurIPS community.

Weaknesses: The experimental evaluation might be criticised as being limited because it is now only for low-dimensional systems. (Meanwhile, I do not think it is a fatal flaw of the paper. Learning 2d or 3d dynamics with some desired properties is indeed a nontrivial problem.)

Correctness: To my knowledge, the technical claims seem valid and correct.

Clarity: Basically the paper is easy to follow. A point that might be unclear is the various definitions of stochastic stability presented. While all the well-known notions are covered in the Background section, what's discussed afterward is only the 2nd mean stability. The authors may want to elaborate on why only the mean stability is treated there and comment on possibility of imposing other types of stability, if possible.

Relation to Prior Work: Relation to prior work is clearly stated.

Reproducibility: Yes

Additional Feedback: Great work! I enjoyed reading the paper. In Examples 1 and 3, readers may be interested in variance of not only learned models (i.e., prediction paths) but also true dynamics. ----- [After rebuttal] Thank you for the rebutal. It was satisfactory. I maintain my evaluation (8), which is originally very positive.


Review 3

Summary and Contributions: In the manuscript, the authors propose methods for learning discrete-time stochastic dynamic models from observed data using DNN. The methods learn the dynamics models using surely stable constraints. Specifically, the proposed methods learn the dynamics to satisfy the Lyapunov stability condition using the Lyapunov function represented by a neural network. The proposed methods have two approaches for each, one exploits convexity of the Lyapunov function, while the other enforces stability through an implicit output layer. Because discrete-time stochastic dynamic models might be important in real-world industrial applications such as control, self-driving vehicles, or anesthesia feedback, this manuscript may deserve to be published in neurips2020. On the other hand, there are some concerns regarding comprehensibility and effectiveness, which are described below, which led me to this judgment.

Strengths: Because discrete-time stochastic dynamic models might be important in real-world industrial applications such as control, self-driving vehicles, or anesthesia feedback, the high estimation accuracy of the proposed method by adding stability constraints to the estimation of noisy discrete-time systems is greatly useful.

Weaknesses: The following four points are considered to be weaknesses. 1. The purpose of the study and the claim of the study is difficult to understand even after reading the abstraction and introduction. For example, the word "deep dynamic models" is not well understood. Also, there is little background explanation of why the authors focus on the discrete-time dynamics model. 2. There is a lack of discussion about the results of the experiments. Please state exactly what you are trying to argue from the results of each experiment. 3. There is a lack of comparison with the methods of previous studies [30]. Isn't it possible to model a stochastic discrete system as a continuous deterministic system and get the same level of performance? 4. The details of the training model and settings, such as the number of samples used for training, are not described. Also, no program is attached. Therefore, reproducibility is not guaranteed.

Correctness: Although γ* in equation (11) is estimated as a constant, the corresponding part of equation (4) varies with x_i. Compared to Eq. (4), the ability to express f(x_i) in Eq. (11) appears to be diminished. Doesn't this poor expressive ability limit the range of application of the method?

Clarity: As I mentioned above, the introduction is written in a way that makes it difficult to understand the purpose of the study and the claim of the study. It is also desirable to have a conceptual diagram of the proposed framework. The quotation of expression numbers, but only the expression numbers are given, making it difficult to understand. How about Eq. (X)?

Relation to Prior Work: As I mentioned above, there is a lack of comparison with the methods of previous studies [30]. It would be needed that the results of previous studies' methods should be described, as well as necessary comparisons of estimation accuracy and computational cost of the methods.

Reproducibility: No

Additional Feedback: As mentioned above, the authors would be better off performing verification to compare the accuracy of the proposed method with the method of previous studies that estimates the Hamiltonian from the wave function without using a DNN. ==== UPDATE AFTER AUTHOR RESPONSE ===== Thank you for your very careful reply. I decide to raise my rate of the manuscript because, including the matter of comparison with previous studies [30] which was my biggest concern, has been resolved.

[Author Response · NeurIPS 2020]



2   We thank the reviewers for their time and effort in providing helpful feedback. We address the comments in order.

3   **Reviewer #1** – No report submitted.

4   **Reviewer #2**   *[Clarity]*   Our supplementary material, Appendix D (ML Reproducibility), includes a detailed description of the network architectures used in our experiments. We used fully connected feedforward networks for both the nominal model $\hat{f}$ and the Lyapunov network $V$. It is worth noting, and we will add this to the paper, that for the nominal model $\hat{f}$ we can use any (parametric/differentiable – for training via backpropagation) representation of the form $x_{t+1} = \hat{f}(x_t, \omega_{t+1})$ to describe the state transitions. Therefore, other architectures such as convolutional neural networks or regularization techniques such as dropout are applicable.   *[Relation to prior work]*   We will elaborate more on the related literature section in the final paper. As for Umlauft and Hirche [39], their approach is constrained to quadratic Lyapunov functions. This constrains the state transition densities that their models can express. Our approach is more general, given the approximation capacity of mixture densities. Moreover, our method applies for non-convex Lyapunov networks and is readily applicable in both deterministic and stochastic settings.

14  **Reviewer #3**   *[Weaknesses]*   We deliberately chose fairly complicated low-dimensional examples: as the reviewer says, these already have legitimate interest and allow for useful visualizations. But the method can readily be applied to high-dimensional states, as our model is a generic feedforward neural network. This allows for applications in model predictive control or reinforcement learning.   *[Clarity]*   We will add some notes on stability concepts and interpretations to the final paper. Here is a preview: 1) The first three definitions align well with deterministic definitions of stability and the corresponding Lyapunov theory, which is our starting point before deriving stability conditions in the stochastic setting; 2) The probabilistic definitions are very strong in the sense that they enforce certain boundedness conditions on the *random variable* $\sup_{t \geq 0} \|x_t\|$, while stability in mean is a path-wise condition. We are then able to exploit properties about the mean/variance of a MDN as well as the Lyapunov network in order to unify Lyapunov theory with 2nd mean stability. In particular, we note that the (conditional) mean dynamics are themselves stochastic processes, which we constrain to be stable in the stochastic Lyapunov sense (there is some discussion of this at the end of section 4 – we force all sample conditional means to decrease in $V$). The 2nd mean condition specifically is simply convenient and could apply for any even-order moment by modifying the proof of Theorem 4.1 slightly.   *[Additional feedback]*   We will add variance info to our figures. Originally, the main drawback was simply how 'busy' the plots become when showing sample paths for both the true system and the learned model.

29  **Reviewer #4**   *[Weaknesses]*   • We can appreciate the confusion that the abstract/introduction may have cause. We apologize for it. Let us clarify. "Deep dynamic models" can be used interchangeably with "DNN-based dynamic models" – DNNs for modeling a dynamical system. The purpose of the paper is to develop architectures with formal guarantees (stochastic stability) about such models. Stability is a critical property in real-world applications. This motivates our focus on stochastic discrete-time systems: even when the system of interest has continuous-time dynamics, we only observe samples $x_t, x_{t+1}, \ldots$ (rather than the functions $x(\cdot)$ and $\dot{x}(\cdot)$), making discrete-time stability the criterion of practical interest. Moreover, uncertainty (such as measurement noise) is unavoidable in practice, which is why we must account for stochasticity while pursuing stability. • We will expand on the discussion of our results in the final paper. Briefly, the first example unifies all the methods in the paper as well as the classical (deterministic) Lyapunov equation with a stochastic analog. The second illustrates how our implicit output layer method provides more flexibility than the convexity-based method for more complex dynamics. The third affirms the usefulness of stability as a 'primitive' in the model over a simple mixture density network. • From the points described in the first bullet, we cannot model a stochastic discrete system as a continuous deterministic system because the previous approach [30] is incompatible with discrete-time observations and does not address uncertainty. We will provide more details discussing the differences between our framework and the one in [30]. • In Appendices B (Training implicit dynamic mode) and D (ML Reproducibility) we provided a full description of the models, data, training, and gradient calculations for backpropagation (for the implicit layer). We will be more clear in the main text of the final paper about what is contained in the appendix. We will also include a GitHub link to our code in the final paper.   *[Correctness]*   Thank you for pointing this out. It is an important point that we will clarify. $\gamma^\star$ in Eq. (11) is a generalization of $\gamma$ in Eq. (4) and is therefore state dependent as well. Eq. (10) expresses the problem for a fixed state, not all states simultaneously.   *[Clarity]*   • The first point is addressed above. • For the conceptual diagram, we felt Figure 1 gave a useful intuitive explanation of our two methods side-by-side. We will add a third sub-figure to illustrate stochastic component as well. • We will use Eq. (X) for clarity as suggested.   *[Relation to prior work]*   We address both of the reviewer's comments in the third bullet under *Weaknesses*. *[Reproducibility]*   We address the reviewer's assessment in the fourth bullet under *Weaknesses*. *[Additional feedback]*   This comment appears to be misplaced, but similar to the one in *Relation to prior work*. We have addressed it above, under *Weaknesses*.

[Meta-Review · NeurIPS 2020]

The reviews were all positive, highlighting that this submission makes good progress on an important problem (stability in learned dynamics). The excellent author feedback clearly answered reviewers' questions and increased their confidence (and mine). Please take the reviewers' suggestions for clarifications into account for the final manuscript, as indicated in the author feedback.